# Assessing the Potential Role of Cats (*Felis catus*) as Generators of Relevant SARS-CoV-2 Lineages during the Pandemic

**DOI:** 10.3390/pathogens12111361

**Published:** 2023-11-16

**Authors:** Ninnet Gomez-Romero, Francisco Javier Basurto-Alcantara, Lauro Velazquez-Salinas

**Affiliations:** 1Comisión México-Estados Unidos para la Prevención de Fiebre Aftosa y Otras Enfermedades Exóticas de los Animales, Carretera Mexico-Toluca Km 15.5 Piso 4 Col. Palo Alto, Cuajimalpa de Morelos, Mexico City 05110, Mexico; ninnet.gomez.i@senasica.gob.mx; 2Departamento de Microbiología e Inmunología, Facultad de Medicina Veterinaria y Zootecnia, Universidad Nacional Autónoma de México, Av. Universidad No. 3000 Col Copilco Universidad, Mexico City 14510, Mexico; basurto@unam.mx; 3Plum Island Animal Disease Center, Agricultural Research Service, United States Department of Agriculture, Greenport, NY 11944, USA; 4National Bio and Agro-Defense Facility (NBAF), Agricultural Research Service, United States Department of Agriculture, Manhattan, KS 66502, USA

**Keywords:** cats, SARS-CoV-2, evolution, variants, phylogenomics

## Abstract

Several questions regarding the evolution of SARS-CoV-2 remain poorly elucidated. One of these questions is the possible evolutionary impact of SARS-CoV-2 after the infection in domestic animals. In this study, we aimed to evaluate the potential role of cats as generators of relevant SARS-CoV-2 lineages during the pandemic. A total of 105 full-length genome viral sequences obtained from naturally infected cats during the pandemic were evaluated by distinct evolutionary algorithms. Analyses were enhanced, including a set of highly related SARS-CoV-2 sequences recovered from human populations. Our results showed the apparent high susceptibility of cats to the infection SARS-CoV-2 compared with other animal species. Evolutionary analyses indicated that the phylogenomic characteristics displayed by cat populations were influenced by the dominance of specific SARS-CoV-2 genetic groups affecting human populations. However, disparate dN/dS rates at some genes between populations recovered from cats and humans suggested that infection in these two species may suggest a different evolutionary constraint for SARS-CoV-2. Interestingly, the branch selection analysis showed evidence of the potential role of natural selection in the emergence of five distinct cat lineages during the pandemic. Although these lineages were apparently irrelevant to public health during the pandemic, our results suggested that additional studies are needed to understand the role of other animal species in the evolution of SARS-CoV-2 during the pandemic.

## 1. Introduction

More than three years after the emergence of COVID-19, SARS-CoV-2 has officially produced, as of 2 November 2023, a total of 771,679,618 disease cases and 6,977,023 confirmed deaths around the world (https://covid19.who.int/, accessed on 2 November 2023). During this time, the scientific community has witnessed the exceptional evolutionary dynamics of this pathogen [1,2]. This situation has promoted the diversification of more than 1300 lineages [3]; some of these lineages are associated with variants of concern (VOC) responsible for the multiple contagion waves produced during the pandemic [4]. On 5 May 2023, the World Health Organization announced that COVID-19 was no longer a public health emergency [5]. However, many aspects regarding the biology of SARS-CoV-2 remain poorly understood, like the potential role of distinct animal species in promoting the emergence of lineages with a likely impact on public health, a situation that may be supported by the rapid evolution of SARS-CoV-2 described after infection of some susceptible animal species like cats and white-tailed deer [6,7,8,9].

SARS-CoV-2 infections were initially documented in humans; however, zooanthroponotic transmission events were also reported as positive cases increased worldwide. Currently, evidence of human-to-animal transmission has been documented in at least 29 domestic and wild animal species from 36 countries (https://www.woah.org/es/documento/sars-cov-2-in-animals-situation-report-22/, accessed on 30 October 2023). Under natural conditions, SARS-CoV-2 infections have been reported in cats, dogs, Syrian hamsters, pet ferrets, mink, raccoon dogs, white-tailed deer, tigers, lions, puma, snow leopards, fishing cats, lynx, hyenas, hippopotamus, otters, gorillas, manatees, black-tailed marmosets, mandrills, coati, and common squirrel monkeys. Likewise, SARS-CoV infections were corroborated under experimental conditions in the companion animals mentioned above. Similarly, experimental infections were performed in further animal species to determine SARS-CoV susceptibility to other animal species, revealing positive infections in laboratory mice, raccoon dogs, red foxes, African green monkeys, Cynomolgus macaque, and deer mice, among others. Conversely, poultry such as chicken, ducks, turkey, quail, and goose are not susceptible to SARS-CoV infection. In contrast, pigs and cattle have extremely low susceptibility to SARS-CoV infection [10,11,12,13,14,15].

Interestingly, in the case of minks, the detection of three mutations (Y453F, F486L, and N501T) located in the receptor binding domain of the spike protein of SARS-CoV-2 may be associated with the adaptation of this virus to this animal species [7]. Furthermore, experimental in vivo co-infections in white-tailed deer revealed the ability of the variant alpha to outcompete the ancestral lineage A, indicating the potential fitness increase in some variants to infect animal species [16].

Due to their close contact with humans, companion animals are important targets for the infection with SARS-CoV-2. This conclusion is consistent with the most updated report (19 September 2023) from the Animal and Plant Health Inspection Service (USDA), showing the increased number of positive cases detected in cats and dogs (https://www.aphis.usda.gov/aphis/dashboards/tableau/sars-dashboard, accessed on 19 September 2023). However, experimental studies have demonstrated unequal levels of susceptibility to SARS-CoV-2 between both species [17,18]. Hence, cats show a higher susceptibility to infection with SARS-CoV-2 than dogs [17]. While in dogs, SARS-CoV-2 is associated with poor replication and lack of transmissibility to naïve dogs [17,18], cats seem to support higher levels of viral replication, with the ability to promote viral transmission among susceptible cats [17,18,19].

Simultaneously to the rapid increase in COVID-19 cases in humans, numerous case studies and research publications have evidenced and reinforced the continuous occurrence of human-to-cat and cat-to-cat SARS-CoV-2 transmission under natural and experimental conditions. The first reported case infection of SARS-CoV-2 was documented in a 15-year-old cat, showing gastrointestinal clinical signs and subsequent respiratory signs one week after the owner was SARS-CoV-2-positive [20]. A second case was reported in Hong Kong from an infected cat with no clinical signs, living with a SARS-CoV-2-positive patient (https://www.info.gov.hk/gia/general/202003/31/P2020033100717.htm, accessed on 31 October 2023). Further, two cases of mild respiratory disease in two cats were described in New York State, where, despite the positive detection of SARS-CoV-2 in the affected cats, the source of infection was only determined in one of the cases related to the infected owner [21]. Subsequently, sporadic detections of SARS-CoV-2 infections in cats showing respiratory and gastrointestinal clinical signs associated with COVID-19-positive owners were reported worldwide [11,22,23,24,25,26]. Thus, these case studies evidence that cats from affected households are at higher risk of infection compared to those without exposure to SARS-CoV-2. On the other hand, fatal or severe clinical manifestations in SARS-CoV-2-infected cats have been seen, mainly related to cat comorbidities [22,27,28,29].

Additionally, serological surveys have been conducted worldwide since the beginning of the pandemic. The seroprevalence rate varies based on the test used to detect anti-SARS-CoV-2 antibodies. In early 2020, in Wuhan, China, an initial study revealed seropositivity in 15 out of 102 cats surveyed [30]. Likewise, in Italy, a study was conducted to assess SARS-CoV-2 infection in companion animals, where 180 cats that tested negative for SARS-CoV-2 revealed the presence of neutralizing antibodies in 5.8% of the evaluated cats [31]. Higher seroprevalence rates for SARS-CoV-2 were observed in France (23.5%) and the USA (17.6%) [32,33]. Conversely, a low percentage of antibody-positive samples was described in Spain (3.51%) [34]. Similarly, during the first wave of SARS-CoV-2, a 0.65% seroprevalence was estimated in Germany [35]; however, this seroprevalence rose to 1.36% by the second wave, revealing that the increase in the number of COVID cases in humans was consistent with the increment of the seroprevalence rate in cats [36].

In general, natural infections reported in cats linked to multiple VOC resulted not only in subclinical outcomes but also in clinical infections, including signs of tiredness, lethargy, fever, ocular and nasal discharge, dyspnea, sneezing, cough, vomiting, diarrhea, anorexia, myocarditis, depression, arching of the back, lack of appetite, and weight loss [37,38,39,40,41]. The disparate clinical outcomes induced by different lineages of SARS-CoV-2 were demonstrated experimentally in cats, showing that the omicron lineage BA.1.1 represents a lower pathogenicity phenotype than the B.1 and Delta (B.1.1.529) lineages [37]. The above-mentioned information suggests that diverse lineages of SARS-CoV-2 represent a different phenotype for cats, emphasizing the relevance of evolutionary analysis to understand the dynamics associated with the possible adaptation of SARS-CoV-2 to cat populations and the consequent emergence of new viral lineages.

In contrast, despite the high susceptibility to SARS-CoV-2, multiple experimental trials in cats resulted in subclinical outcomes [18,37,42], highlighting the relevance of cats transmitting this virus in conditions of inapparent signs of disease. This situation may represent a potential risk factor for cat owners.

Interestingly, experimentally, evolutionary analyses in cats showed the rapid evolution of SARS-CoV-2 after infection in this animal species [7,8], suggesting the potential relevance of cats as a generator of new SARS-CoV-2 variants during the pandemic. Furthermore, phylogenetic analyses comparing sequences obtained from natural infections with SARS-CoV-2 in cats and human populations indicated the high susceptibility of cats to the infection with lineages affecting humans, highlighting the unidirectionality of the transmission events between both species (zooanthroponotic transmission) [7].

Based on the above, our study aimed to evaluate the evolutionary dynamics of diverse SARS-CoV-2 lineages related to natural infections in cat populations. For this purpose, we used a combination of multiple evolutionary algorithms to evaluate a set of full-length viral sequences recovered from natural infections in cats during the pandemic. Moreover, including a group of highly related viral sequences obtained from human infections, we developed an interesting evolutionary model to test the possible role of natural selection in the emergence of SARS-CoV-2 due to replication in cat populations. Our results are discussed regarding the relevance of cats as generators of new SARS-CoV-2 lineages and the potential impact of these on public health.

## 2. Materials and Methods

### 2.1. Viral Sequences and Metadata Information

A total of 105 complete viral genomes from cats naturally infected with SARS-CoV-2 during the pandemic were retrieved from the GISAID database [43]. The average length of sequences was 29,826 nucleotides. These 105 sequences were chosen from a total of 168 full-length available sequences under the GISAID criteria of complete (considered genomes >29,000 nt) and high coverage (only entries with <1% nt and <0.05% unique amino acid mutations not seen in other sequences in the database, as well as no insertion/deletion unless verified by the submitter). Specific information about each sequence is provided in Appendix A. Additionally, 117 complete viral genomes from human cases showing the highest levels of identity against cat SARS-CoV-2 genomes were obtained from the GenBank database.

Metadata information (2515 reports) available on 16 April 2023 in the GISAID database [43] regarding the description of reports of SARS-CoV-2 infections in cats and nine other animal species was considered for this study. It included: cat (*Felis catus* = 179 reports), dog (*Canis lupus familiaris* = 125 reports), white-tailed deer (*Odocoileus virginianus* = 583 reports), lion (*Panthera leo* = 78 reports), tiger (*Panthera tigris* = 43 reports), Western gorilla (*Gorilla gorilla* = 17 reports), American mink (*Neogale vison* = 1391 reports), house mouse (*Mus musculus* = 50 reports), Syrian hamster (*Mesocricetus auratus* = 30 reports), and pangolin (*Manis javanica* = 19 reports). Reports were analyzed to create a compiled file containing the overall number of SARS-CoV-2 pangolin lineages and the genetic clade (GISAID classification) comprising these lineages implicated in the infection of multiple species. As a result, a total of 155 pangolin lineages associated with ten genetic groups (S, L, V, O, G, GR, GH, GK, GV, and GRA) were identified. After that, the proportion of reports from each species associated with these lineages was determined (Appendix A).

### 2.2. Hierarchical Cluster Analysis

Hierarchical cluster analysis (HCA) was conducted using the information presented in Appendix A to assess a comparison between SARS-CoV-2 viral lineages recovered from cats and other animal species. This analysis was performed using the predictive analytic software JMP^®^ Pro version 16.0.0. Generally, HCA uses a matrix of N × M dimensions, where N corresponds to the number of animal species included in the analysis and M is the proportion of reports associated with the infection of SARS-CoV-2 by specific pangolin lineages. Clusters were defined by the Ward linkage method. Statistical significance between clusters was conducted by analysis of variance (ANOVA) and supported by Tukey’s honest significance test (*p* = 0.05) to define the most prevalent pangolin lineages implicated in the infection of different animal species included in this study.

### 2.3. Correlation between Genetic Groups Affecting Humans and Cat Populations during the Pandemic

The coefficient of determination (R^2^) was calculated to assess the correlation between the dominance of specific genetic groups of SARS-CoV-2 in human populations and the identification of cat infections during specific pandemic years. For this purpose, information regarding the number of sequences in humans associated with specific genetic groups from 2019 to 2022 was retrieved from the GISAID database on 16 April 2023. Each year, the proportion of viral sequences related to specific genetic groups was calculated as follows: Proportion of specific genetic group = total number of sequences associated with a specific genetic group/total number of sequences associated with multiple genetic groups reported at one particular year. Similar calculations were conducted on cat populations to determine the proportion of different genetic groups identified at specific years. Coefficient of determination analyses were performed in GraphPad Prism 9.5.0 software, considering a *p* < 0.05 as a significant correlation between the genetic groups associated with human and cat populations at any particular year. The idea of using genetic groups of SARS-CoV-2 lineages for this analysis was to obtain an alternative perspective from the results obtained from the hierarchical cluster analysis using pangolin lineages.

### 2.4. Phylogenetic Analysis

The phylogenetic relationship between multiple viral sequences recovered from naturally infected clinical cases in cats and humans was assessed by the maximum likelihood method using the general time reversible model (GTR), considering gamma distribution and invariable sites. The use of this model was supported by the lowest Bayesian information criterion scores (BIC). Analyses were conducted in the molecular evolutionary genetic analysis software MEGA version 10.2.5 [44]. Additionally, this software was used to perform the pairwise distance analysis included in this study.

### 2.5. Evolutionary Analysis

To assess the evolutionary dynamics among SARS-CoV-2 viral lineages recovered from cats, we used the evolutionary algorithms Fixed Effects Likelihood (FEL) [45,46] and Mixed Effects Model of Evolution (MEME). Both algorithms detect sites under negative or positive selection, acting in a pervasive (MEME and FEL) and episodic (MEME) manner by inferring rates of synonymous and nonsynonymous substitutions in a codon-base phylogenetic framework. Moreover, the presence of viral lineages potentially emerging because of natural selection was evaluated by the algorithm Adaptive Branch-Site Random Effects Likelihood (aBSREL) [47]. aBSREL uses the likelihood ratio test statistic for the selection (LRT) of relevant branches in the tree and the Holm–Bonferroni method for the correction p-values. Subsequently, the algorithm Branch-Site Unrestricted Statistical Test for Episodic Diversification (BUSTED) [48] was used to identify possible sites under positive selection in the branches detected by aBSREL. BUSTED is a likelihood ratio algorithm to test for evidence of diversifying selection affecting some positions in the alignment along some branches of the tree [49].

## 3. Results

### 3.1. Overview of the SARS-CoV-2 Lineages Isolated from Cat Infections around the World

A description of the cat reports included in this study is provided in Figure 1. Overall, available reports in cats at the GISAID database document infections in cats on multiple continents (Figure 1A). The highest percentage (60.33%) of reports accounted for just three countries (US, South Korea, and Switzerland); out of these, the US represents the country with the highest number of reports (n = 62). Conversely, the rest of the reports (39.67%) were associated with 26 countries (Figure 1A), showing disparate information regarding the infection of cats with SARS-CoV-2 worldwide. Interestingly, regarding gender, a significant association (*p*-0.022) was revealed by the Fisher exact test between the number of clinical reports and the gender of cats, suggesting that male cats may have a higher predisposition for SARS-CoV-2 infection. Contrariwise, no significant association (*p*-0.33) was found between the number of reports and the age of cats (Figure 1B,C).

### 3.2. Assessing the Prevalence of SARS-CoV-2 Lineages in Cats and Other Animal Species during the Pandemic

Hierarchical cluster analysis was conducted to obtain a general perspective on the prevalence of distinct SARS-CoV-2 pangolin lineages that infected cats and nine other animal species during the pandemic. Cats had the highest number of lineages associated with clinical infections (n = 73), followed by dogs and white-tailed deer, with 56 and 48, respectively (Figure 2). Hierarchical cluster analysis grouped lineages into two different clusters. While cluster 1 comprised most of the lineages representing single reports in cats, cluster 2 included a specific group of just lineages associated with multiple clinical reports. Interestingly, this situation was consistent for cats and most animal species included in the analysis, suggesting the potential relevance of these lineages to infect animal species during the pandemic (Figure 2). The preponderance of lineages associated with cluster 2 was assessed by analysis of variance (*p* > 0.001) and confirmed by Student’s *t*-test (*p* > 0.001). Furthermore, more prevalent lineages grouped in cluster 2 were associated with the genetic groups S, G, GV, GRA, GH, GK, and GR. From these, genetic group GK comprised the highest number of lineages (AY.103, AY.3, AY.69, AY.44, and AY.20), highlighting the potential relevance of this genetic group to produce animal infections during the pandemic.

Additionally, no correlation was found between the prevalence of these lineages in human populations and the number of reports in cats and other species (Figure 2). For example, in the case of cats, except B.1.1.7 (Alpha variant) representing 7.66% of the total number of SARS-CoV-2 sequences reported in the GISAID database and the second most prevalent in cats (7.79%), B.1, the most prevalent lineage reported in cats associated with 10.38% of the reports, represented just 0.88% of the sequences in GISAID related to human infections. In this context, the most prevalent lineages in humans (BA.2, BA.1.1, AY.4, BA.1, and AY.43) were associated with main cluster 1 (Figure 2).

Clinical reports showing cat deaths were related to single lineages, including B.1.526, P.1, B.1.369, B.1.36.35, B.1.234, and B.1.2 (the only one included in cluster 2) (Figure 2).

### 3.3. Evolutionary Dynamics of Viral Lineages Affecting Cat Populations

Once we obtained a general overview of the most prevalent pangolin lineages and the genetic groups of SARS-CoV-2 among cats and other animal species, we attempted to obtain more insights into the potential factors shaping the evolutionary dynamics of this pathogen in cat populations. For this purpose, we analyzed if there was a correlation between the dominance of specific genetic groups in the infections in human populations and the genetic clades associated with the infections in cat populations.

Generally, between 2020 and 2022, we found a positive correlation between the genetic groups affecting humans and cat populations, indicating that viral genetic groups affecting cats during the pandemic were highly influenced by the circulation dynamics of these groups in humans. The latest can be exemplified by the dominance of the GK and GRA genetic groups in human populations and the high correlation (R^2^ = 0.99) in the prevalence of these genetic groups in cat infections during 2022 (Figure 3A). GH and GK genetic groups were associated with most clinical reports evaluated in this study (Figure 3B).

Subsequently, to obtain more insights into the phenotypic characteristics and the evolutionary dynamics of viral lineages affecting cats during the pandemic, we conducted a phylogenomic analysis using a representative set of 105 sequences (Figure 4). This number was defined based on sequence quality (see Section 2). Thus, we favored sequences that preserved the integrity of multiple viral genes. The pairwise analysis showed an identity between 99.62% and 100% (~99.86%) and 99.20% and 100% (~99.72%) at the nucleotide and amino acid levels, respectively, indicating the high conservation among SARS-CoV-2 viral lineages affecting cat populations. Furthermore, 100% identity was found between some viral sequences from genetic groups GH (9 B.1.497 GH 21-11 B.1.497 GH 21 and 55 B.1 GH 20-56 B.1 GH 20), GR (72 B.1.1.254 GR 20-73 B.1.254 GR 20 and 87 B.1.1.298 GR 20-89 B.1.1.298 GR 20), and GK (100 AY.3 GK 21-101 AY.3 GK 21). Considering our results regarding the differences in the prevalence of different viral lineages during the infection in cats (Figure 2), our phylogenetic inference showed the existence of multiple events of divergence within some of the most prevalent lineages (B.1.497, B.1, B.1,2, AY.69, AY.103, AY.3, B.1.1.7) (Figure 4), indicating that cats were infected with potentially distinct subtypes of each lineage. In addition, dN/dS rates equal to 0.6584 obtained by FEL analysis indicated that purifying selection was the main force driving the evolution of SARS-CoV-2 among cat populations. However, dN/dS calculations within different lineages evidence that purifying selection is dissimilarly acting among them (Figure 4). Thus, it was possible to observe the contrasting dN/dS rates between B.1.497 (dN/dS = 0.905) and AY.3 (dN/dS = 0.346), suggesting that lineages infecting cats were subjected to different evolutionary pressures during the pandemic.

An additional phenotypic characteristic of viral lineages affecting cats during the pandemic was the presence of premature stop codons at the ORF8 gene at codon positions 27 and 64 of 9 viruses (Figure 4), suggesting that ORF8 is not an essential protein for the infection in cats.

Using the algorithm MEME, ten codon sites were identified along the genome, involving six different genes, with the S gene accounting for the highest number of codons under positive selection (n = 4) (Figure 5A). On the other hand, the most frequent substitutions in the population were mutations at NSP2-85 (n = 28), NSP4-492 (n = 22), and S-452 (n = 32) gene codons. Furthermore, most of the codons identified under positive selection were wildly spread among diverse genetic groups, even when some were present in the population at lower frequencies (NSP16-216, S-95, S-477, S-501, and N-13). When we evaluated the dataset by the algorithm FEL, we identified codons NSP4-492, NSP16-216, S-95, S-452, S-477, and N-13 under positive selection (*p*-value 0.1), suggesting that these codons were evolving under pervasive diversifying selection, hence highlighting the relevance of these codons among different genetic groups. Conversely, codons identified under positive selection just by MEME analysis (NSP2-85, NSP2-280, S-501, and N-3) might be associated with an evolutionary pattern of episodic selection.

Once we identified potentially relevant codon positions during the infection of SARS-CoV-2 in cat populations, we analyzed the evolutionary dynamics of those positions during infections of this pathogen in human populations (Figure 5B). For this purpose, we attempted to recreate a dataset of viral sequences associated with human infections highly related to viral sequences recovered from cats. We conducted a BLAST analysis, using each of the 105 cat sequences used in this study as a query. As a result, a total of 117 representative viral sequences recovered from human infections were obtained for this analysis. Phylogenetic analysis using this new dataset evidenced the close genetic relationship between both viral populations (Figure 6). No evidence of specific SARS-CoV-2 host adaptation was observed. However, based on the branch topology and the ancestral relationship between cat and human lineages displayed in the tree, it is possible to predict the existence of multiple divergence events that occurred during the infection of SARS-CoV-2 in cats, suggesting that human–cat transmission events promoted the evolution of SARS-CoV-2.

Furthermore, MEME analysis showed that all positions identified in cat populations (Figure 5A) appeared under positive selection in human populations (Figure 5B). Overall, we found that mutations associated with these codons have been found in the SARS-CoV-2 lineages, affecting humans in numerous countries globally (Figure 5B). When we assessed the Pearson correlation between the allele frequency found in cats at different codons under positive selection and the frequency of these alleles in viral sequences recovered from humans at the GISAID database (Figure 5B), we found a significant positive correlation (*p*-value = 0.0159; r = 0.5955) in the frequency between both populations, indicating that the viral circulation dynamics of SARS-CoV-2 in the human population influenced the phenotypic characteristics of lineages affecting cats. Suitable examples of the latter can be observed in codon NSP2-280, where the mutation AAC-TAC (present in a single viral sequence in cats) represents just 0.53% of the mutations reported in the GISAID database in the human population, and in the polymorphic codon sites NSP16-216 and S-452, where codons with low-frequency substitutions appeared at frequencies lower than 1% in human populations (Figure 5B).

In contrast, when we compared dN/dS rates between multiple gene segments from viral populations recovered from humans and cats, we observed a disparity between both populations in the dN/dS rates from genes ORF3a, E, ORF7b, and ORF10 (Figure 5C), suggesting that replication of SARS-CoV-2 in these hosts may represent a dissimilar selective environment.

### 3.4. Evaluating the Hypothesis about Independent Evolution of SARS-CoV-2 in Cat Populations

We tested the hypothesis about the potential independent evolution of SARS-CoV-2 in cat populations. To this end, we used data comprising 105 viral sequences recovered from naturally infected cats and a set of 117 highly related viral sequences recovered from human infections. The algorithm aBSREL evaluated this dataset in an attempt to discover potential cat lineages evolving because of natural selection. Interestingly, five different cat lineages were found under positive selection (Figure 7A). Three of these lineages were associated with the pangolin lineages B.1 (55 B.1 GH 20), AY.69 (22 AY.69 GK 21 and 29 AY.69 GK 21), some of the most prevalent lineages in cats. The other two cat lineages were associated with the pangolin lineages BA.1.1 (34 BA.1.1 GRA 22) and B.1.1.298 (88 B.1.1.298 GR 20). Phylogenetic analysis showed the close genetic relationship of these lineages with viral lineages associated with human populations, with a nucleotide identity calculated between 99.94% and 99.99% (Figure 7B–E). Overall, with the exception of cat lineage 34 BA.1.1 GRA 22, where a total of 12 SNPs were identified among seven different genes (Figure 7C) between this lineage and the closet human lineage, the rest of the lineages between 1 and 3 SNPs were identified in lineages: 22 AY.69 GK 21, 29 AY.69 GK 21, 55 B.1 GH 20, and 88 B.1.1.298 GR 20, impacting a minimal number of genes (Figure 7B,D,E).

Finally, to identify potential codon sites linked to the emergence of cat lineages, we evaluated previously relevant branches identified by aBSREL using the algorithm BUSTED. Evidence ratios (site level likelihood ratios) for ω > 1 were found in eight different codon sites among incident branches of identified cat lineages (Figure 7F). Four were identified in the lineage 34 BA.1.1 GRA 22 from these sites, while single sites were identified in the remaining lineages. Subsequently, we assessed the evolutionary significance of these sites, using specialized algorithms to identify natural selection at codon sites (FEL and MEME). There was evidence of diversified selection in just two sites associated with isolation 34 BA.1.1 GRA 22 (NSP6 codon 260 and S codon 211) (Figure 7F). In both cases, FEL identified evidence of diversifying selection in the external nodes, suggesting that these codon sites did not give an adaptive advantage to SARS-CoV-2 at the cat population level.

Afterward, to assess the impact during the pandemic of the alleles present in codon sites potentially linked to the emergence of cat lineages, we evaluated the frequency of these alleles at specific codon sites in the GISAID database. Our results indicated that in all cases, the occurrence of the alleles associated with multiple cat lineages was present at a low frequency in the GISAID database (Figure 7F). The latter suggests the lack of relevance of these alleles in promoting the adaptation of dominant SARS-CoV-2 lineages during the pandemic. Conversely, at these codon sites, alleles present in viral sequences isolated from humans, including the ones associated with the possible ancestral lineages predicted in our model, matched with the most frequent allele for this codon site described in the GISAID database. The only exception was the allele present in the predicted ancestral sequence for cat lineage 34 BA.1.1 GRA 22, where at gene S codon 211, the human isolate has the allele AAA that showed an overall frequency at this codon position of 0.01% (Figure 7F).

## 4. Discussion

At this point, and consistently with the view presented in previous publications [2,4,7,8,50], we consider that deciphering the evolutionary dynamics of SARS-CoV-2 appears to be an imperative task to understand the future impact of SARS-CoV-2 in the world. As the main host of this pathogen, during the pandemic, multiple research subjects involving evolutionary aspects of COVID-19 have focused mainly on understanding this disease from the human standpoint [2,4,50]. However, as mentioned before, multiple animal species were affected by SARS-CoV-2 during the pandemic [6,7,8,9], which shows this viral agent’s complex biology. Currently, limited information is known about the potential role of animal species in the epidemiological triad of this disease. In this sense, a recent study conducted on white-tailed deer in the US showed the relevance of this animal species as a potential reservoir of SARS-CoV-2, promoting not only the evolution of this virus but also probably spillovers to human populations [51], stressing the preponderance of some animal species in the evolution of SARS-CoV-2, and the potential consequences for public health.

Herein, we focused on SARS-CoV-2 infections in domestic cats. We considered cats a potentially relevant animal species for the evolution of SARS-CoV-2 based on their high susceptibility to this virus and its ability to transmit the infection to susceptible cats [17,18,19]. In this context, to obtain more insights about the conceivable consequences of the evolution of SARS-CoV-2 produced by natural infections in cat populations, we aimed to describe the evolutionary dynamics of viral lineages affecting cat populations during the pandemic. The results of our study provide a different but complementary perspective to previous studies [7,8], supporting the potential role of cats as generators of new variants of SARS-CoV-2.

Our study has diverse limitations to be considered in interpreting the results presented here. The most important limitation is the reduced number of available full-length sequences obtained from naturally infected cats during the pandemic, representing equally multiple geographical regions worldwide. This fact may have hindered the detection of additional codon sites under positive selection and lineages evolving from cat populations due to natural selection. Considering that almost half of the sequences used in this study were collected from the US, we could not determine potential impacts on the evolutionary dynamics of SARS-CoV-2 in cats based on region-dependent factors. It highlights the imperative necessity to increase surveillance and the consequent production of viral sequences from cats and other animal species worldwide. Moreover, it is crucial to consider that the results from this study were obtained entirely by in silico approaches, warning that experimental evidence is necessary to validate the relevance of our predictions. Therefore, sequencing mistakes affecting the quality of the viral sequences used in this study are another factor to consider in interpreting the results from this study.

To assess the susceptibility of cats to the infection with SARS-CoV-2 in nature, we first compared clinical reports available in the GISAID database between cats and multiple other animal species. Our results indicated that, compared with other animal species, the infections in cats were associated with a higher number of viral lineages (Pangolin lineage classification), indicating the increased susceptibility of this animal species in nature. However, although the results of our analysis might have been biased by the limited available metadata information from cats and other animal species, our results were consistent with the differences in susceptibility experimentally evidenced between cats and dogs [17,18], supporting the view that cats are highly susceptible animal species with a potential role for the transmission of SARS-CoV-2 in nature [52,53]. Furthermore, we found a significant association between the number of clinical reports and the gender of the cats. This result was consistent with a previous study conducted in Germany, suggesting an increased probability of testing positive for SARS-CoV-2 in male cats [54].

Interestingly, in humans, male sex has been described as a predominant risk factor for severe course and poor prognosis during SARS-CoV-2 infections. However, at this point, there is no specific explanation for the disparate susceptibility between genders [55]. In the case of cats, future pathogenesis studies are needed to confirm the existence of potential differences in the susceptibility to SARS-CoV-2 between both genders.

By hierarchical cluster analysis, we identified specific lineages of SARS-CoV-2 associated with the highest prevalence of cases in different animal species. Remarkably, the detection of these lineages was coherent among cats and other multiple animal species, supporting the accuracy of our results and thus highlighting the potential ability of these lineages to infect other species other than humans efficiently. In this context, our last assertion may be supported by our result showing that, except for lineage B.1.1.7 (alpha), most of the highly prevalent lineages found infecting multiple animal species did not correspond with the high prevalence of these lineages in human populations [56]. In this sense, since the metadata information used for this analysis was retrieved from reports not strategically collected for this study, it is important to consider that a variable like the sample collection date might have represented an important source of bias by either impairing the levels of prevalence of specific lineages or preventing the detection of lineages distinct to the ones described in our study.

Contrariwise, our analysis identified some lineages that may have more affinity for specific animal species. Although this result could have been highly influenced by the reduced metadata information about different species, it may be consistent with the experimental evidence showing the disparate levels of pathogenicity associated with diverse SARS-CoV-2 lineages in cats [37].

Consequently, we focused on understanding if the circulation dynamics of SARS-CoV-2 in cats were influenced by the dynamics in human populations. We conduct this analysis using the GISAID genetic group classification, which congregates multiple pangolin SARS-CoV-2 lineages in phylogenetic clusters based on the statistical distribution of the genome distances among lineages [57]. The positive correlation found in this study between the circulation dynamics of different SARS-CoV-2 genetic groups in cats and human populations during the pandemic suggested that natural infections in cats were influenced by the circulation dynamics of SARS-CoV-2 in human populations. The latter evidences the unlikely role of cats as reservoirs of SARS-CoV-2 in nature. On the other hand, despite the discrepancy between these and our previous results that showed a lack of association between the most prevalent pangolin lineages in cats and the dominant ones in human populations (hierarchical cluster analysis, Figure 2), it makes it possible to suggest that regardless of whether infections in cats were influenced by the dominant genetic groups in human populations (coefficient of determination, Figure 3), specific pangolin lineages within these groups might have had more affinity to infect cat populations. This conclusion may be supported by a previous experimental study showing disparate levels of pathogenicity induced by different SARS-CoV-2 lineages in cats [37]. This condition may explain the discrepancy between both analyses.

The evaluation of the phylogenomics of viral lineages that affected cat populations throughout the pandemic denoted the existence of multiple events of divergence within different lineages during the infection in cat populations. The evolutionary significance of these events was more evident when we reconstructed the phylogenetic analysis along with a set of highly related viral genomes recovered from human populations. Thus, two different conclusions can be drawn from this analysis: (i) Consistent with a previous study evaluating the phylogenetic relationship between SARS-CoV-2 lineages associated with the delta variant recovered from cats and human populations [7], our analysis did not show any topological host pattern association among different lineages recovered from both species, indicating the lack of host-specific adaptation patterns; (ii) despite the lack of host-specific adaptation patterns, the branch topology observed (denoting multiple events of divergence) among diverse viruses within the same lineage recovered from both hosts evidenced the evolution of SARS-CoV-2 in cat populations.

Additionally, to obtain more insights into the evolutionary dynamics of SARS-CoV-2 in cat populations, our dataset was evaluated by multiple evolutionary tests previously used in SARS-CoV-2 studies [50]. We aimed to understand if the multiple divergence events observed in our phylogenetic analysis might be linked to natural selection or genetic drift. The overall dN/dS rate < 1 calculated in the viral lineages from the cat population indicated that negative selection was the main force shaping the evolution of SARS-CoV-2 in cats. This result suggests that in nature, human–cat transmission events were characterized by a strong evolutionary constraint in cats, favoring the preservation of the viral phenotypes circulating in humans. Our result was consistent with an experimental study in cats, showing the relevance of purifying selection during the infection of SARS-CoV-2 in this animal species [58]. However, when we calculated dN/dS ratios within different lineages, we observed disparate levels of purifying selection among them, suggesting that different lineages were subjected to different evolutionary constraints. This situation may be consistent with the phenotypic differences depicted in diverse SARS-CoV-2 lineages during experimental infections in cats [37].

In evaluating the presence of positive selection at different codon sites of the viral genomes recovered from cat populations, we described multiple sites under positive selection. However, when we analyzed the dataset from human populations, we found that all these sites were already selected because of the evolution of SARS-CoV-2 in human populations, supporting our last statement about the essential role of purifying selection during the evolution of SARS-CoV-2 in cat populations. In this context, some of the codon sites predicted under positive selection in both populations, including NSP4-492 (increased infectivity and evasion of the immune response [59], S-95 (increased infectivity and transmissibility) [60], S-452 (decreased sensitivity to monoclonal antibodies) [61,62,63], S-477 (decreased sensitivity to monoclonal antibodies) [64,65], S-501 (decreased sensitivity to monoclonal antibodies, enhance infection and transmission) [66], and N-13 (Affect CD8+ response) [67], have shown their biological relevance for the immune host evasion produced by SARS-CoV-2, suggesting that these positions may provide adaptive advantages for this virus during the infection in both hosts. However, despite the failure to infer putative specific codon sites evolving under positive selection exclusively in the viral population recovered from cats, our comparison among genes between viral populations recovered from both hosts suggested disparate evolutionary rates between viruses recovered from different hosts, indicating the possible different evolutionary contrast imposed by humans and cats during the infection. In this sense, further experiments are required to explain the increased levels of positive selection found in the E and ORF10 genes in cat populations. Based on previous publications, it may be possible to speculate about probable repercussions in controlling the immune host response [68,69,70].

Additionally, we found lineages related to the B.1.1.7 lineage carrying a premature stop codon in the genome that inactivates the ORF8 protein with the ability to infect cat populations. This situation has been previously described in human [71], pangolin, and vison populations [72]. Interestingly, humans infected with lineages that contain truncated versions of ORF8 showed less severe symptoms, thus potentially increasing the opportunities for virus spread from apparent subclinical patients [73]. Experiments in mice have shown the relevant role of ORF8 in the pathogenesis of SARS-CoV-2 in this species. More experimental studies are needed to understand the potential role of ORF8 in the pathogenesis of SARS-CoV-2 in cats.

Finally, according to the algorithm used in aBSREL, our results showed evidence of the potential action of natural selection in the emergence of SARS-CoV-2 lineages due to replication in cat populations. However, considering the minimal number of cat lineages found evolving by the action of natural selection and the topology showing the divergence events that occurred between human and cat lineages, it is possible to suggest that genetic drift is another significant force shaping the evolution of SARS-CoV-2 in cats. This result was consistent with a previous experimental study in cats [58]. Also, we found that specific mutations linked to the identified cat lineages under positive selection had minimal impact on human populations, suggesting the unconnected role of cats as generators of relevant SARS-CoV-2 variants during the pandemic.

In summary, our study shows a perspective on the evolution of SARS-CoV-2 in cat populations during the pandemic. Our findings were consistent with previous publications using experimental approaches in cats, indicating the rapid evolution of SARS-CoV-2 in this domestic animal, with purifying selection and genetic drift as the main evolutionary forces acting on this virus during the infection in cats. Although the main conclusion of our study is the possible lack of relevance of cats as generators of relevant variants for public health, considering the high susceptibility of cats to the infection of SARS-CoV-2, it is important to warn about the possible role of cats in the transmission of this pathogen. Considering the phenotypic differences observed experimentally among SARS-CoV-2 lineages in cats [14], it is essential to state that our results may be regarded as inconclusive based on the limited dataset used in this study. In this sense, more studies are needed to understand the potential role of cats in the generation of relevant SARS-CoV-2 lineages for public health, especially in the context of the pangolin lineages XBB.1.5, XBB.1.16, EG.5, DV.7, XBB, XBB.1.9.1, XBB.1.9.2, XBB.2.3, and BA.2.86 currently dominating the infections in human populations (https://www.who.int/activities/tracking-SARS-CoV-2-variants, accessed on 8 November 2023).

## Figures and Tables

**Figure 1 pathogens-12-01361-f001:**
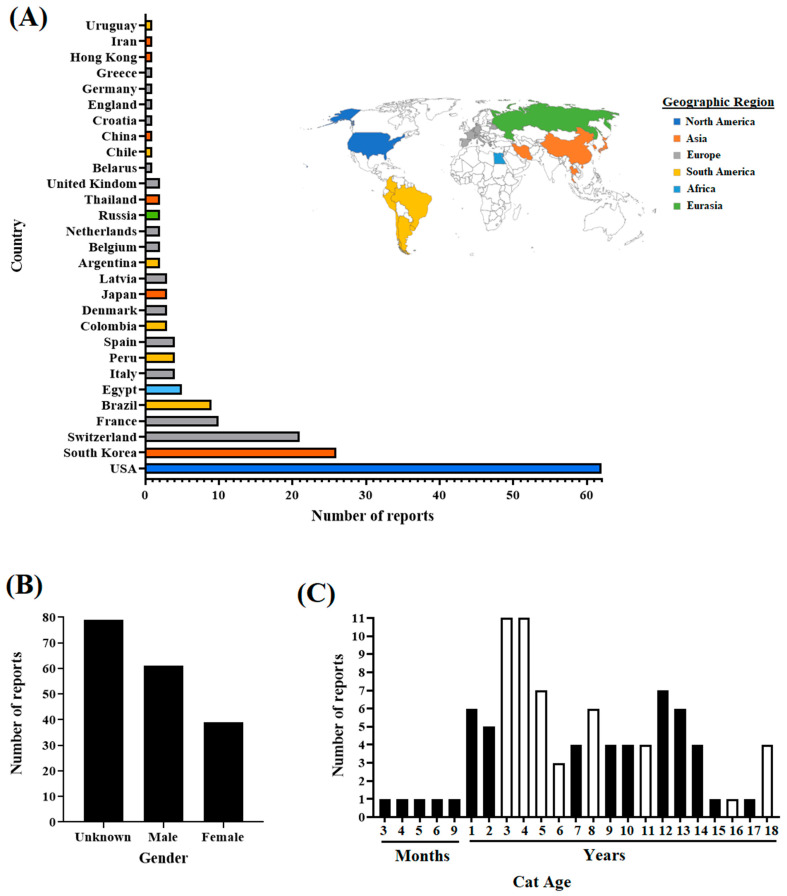
General overview of the metadata information associated with the SARS-CoV-2 sequences from cats used in this study. A total of 179 reports related to cat sequences submitted to the GISAID database were considered. Reports included sequences submitted in 2020 (65 sequences), 2021 (103 sequences), and 2022 (11 sequences). (**A**) Viral sequences recovered from cat infections are described in 29 countries, covering multiple geographic regions. (**B**) The gender of the cats was reported in half of the reports (n = 90), of which 67.77% came from male cats. (**C**) The age of the cats ranged from 3 months to up to 18 years. Opened bars represent ages where cat deaths were reported (one report for each age). In this case, age was recorded in just 84 reports.

**Figure 2 pathogens-12-01361-f002:**
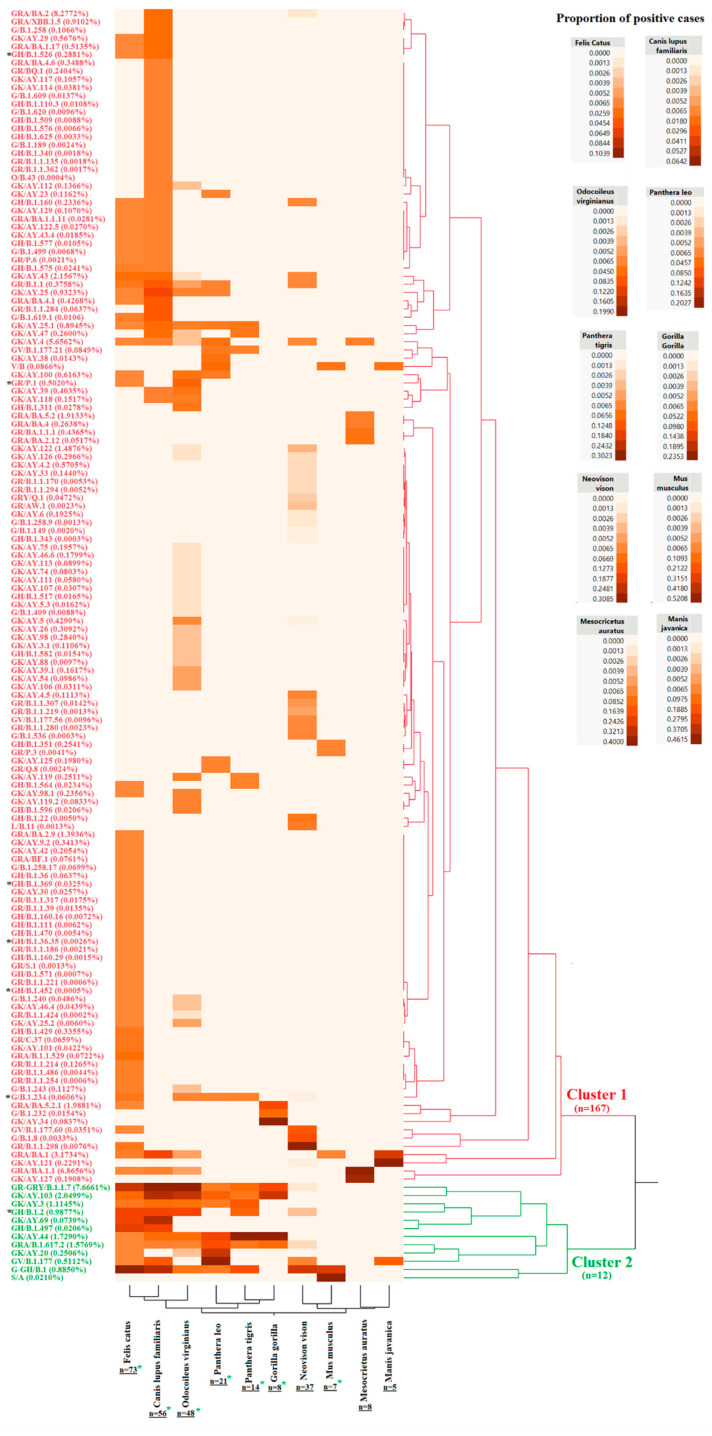
SARS-CoV-2 viral lineages infect cats (*Felis catus*) and other animal species. As a result of the analysis of a total of 2515 reports associated with infections of SARS-CoV-2 in multiple animal species, 155 pangolin lineages were identified as responsible for these infections. The proportion of cases within species related to these lineages was analyzed by hierarchical cluster analysis to assess the relationship between lineages that affected *Felis catus* and other animal species during the pandemic. These species include the dog (*Canis lupus familiaris*), white-tailed deer (*Odocoileus virginianus*), lion (*Panthera leo*), tiger (*Panthera tigris*), Western gorilla (*Gorilla gorilla*), American mink (*Neogale vison*), house mouse (*Mus musculus*), Syrian hamster (*Mesocricetus auratus*), and pangolin (*Manis javanica*). Two main clusters were identified in the analysis. Cluster 2 (green) comprises the 12 more prevalent lineages among different species. The Y-left axe shows the SARS-CoV-2 pangolin lineages included in the analysis. Identification of these lineages included the genetic group/pangolin lineage. The number in the parenthesis next to each lineage indicates the proportion of this specific lineage among the total number of sequences reported in the GISAID database. Black asterisks reflect lineages where mortality was reported in cats. Numbers below animal species in the X-axis show the number of lineages reported during the infection in different species. At the same time, green asterisks indicate species where the prevalence of positive cases was statistically significantly higher in the lineages associated with cluster 2. The spectrum of colors ranges from cream to brown and reflects a low to high proportion of the reports related to specific pangolin lineages, respectively.

**Figure 3 pathogens-12-01361-f003:**
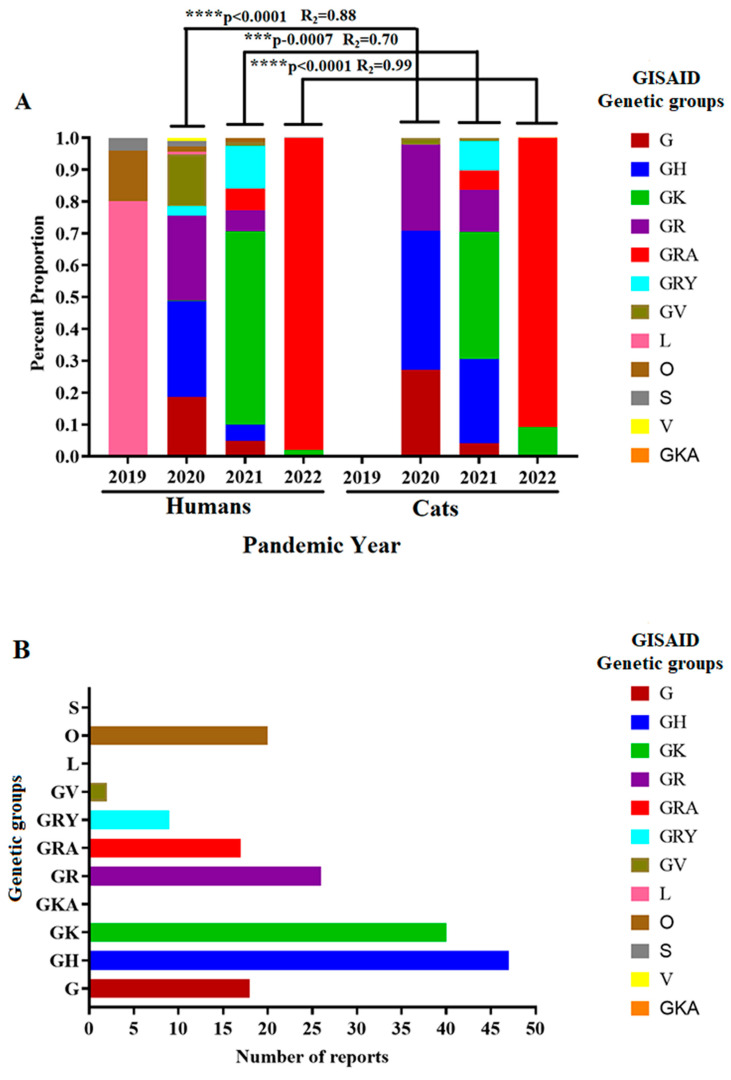
Circulation dynamics of the SARS-CoV-2 lineages during the pandemic. (**A**) The graphic shows the circulation dynamics of different genetic groups of SARS-CoV-2 (based on the GISAID classification) described during the pandemic and their relationship with those circulating in cat populations. The coefficient of determination (R^2^) analysis is presented between the circulation of specific genetic groups affecting humans and cat populations during the same pandemic year. Asterisks indicate significant correlations. (**B**) The graphic shows the overall number of clinical reports in cat populations associated with specific genetic groups of SARS-CoV-2 (GISAID classification).

**Figure 4 pathogens-12-01361-f004:**
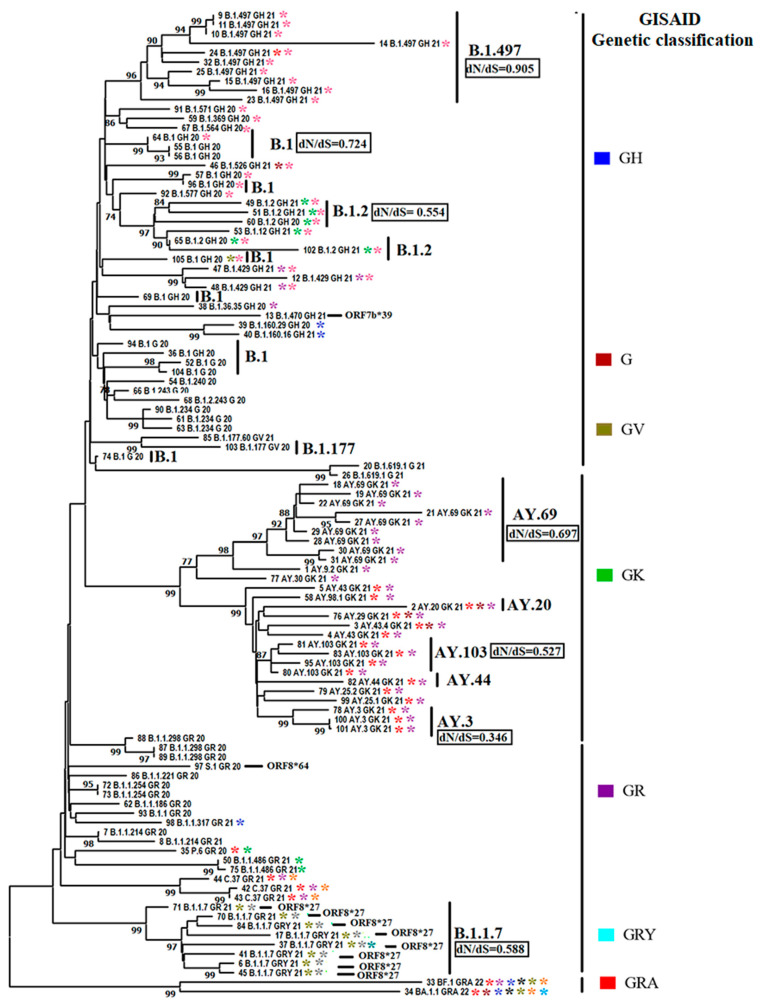
Phylogenomics of SARS-CoV-2 lineages affecting cats. A phylogenetic analysis was conducted using 105 full-length viral sequences retrieved from the GISAID database (specific details about the sequences can be found in Appendix A). The analysis was performed using the maximum likelihood method and the general reversible model. In the tree, labels B.1.497, B.1, B.1.2, B.1.177, AY.69, AY.20, AY.103, AY.44, AY.3, and B.1.1.7 identify clusters associated with the most prevalent SARS-CoV-2 pangolin lineages related to cat infections. The right pangolin lineages are grouped in the context of GISAID genetic group classification. The dN/dS ratios are indicated for the most prevalent cat lineages. Calculations were conducted using the FEL algorithm. Asterisks of different colors reflect specific codon sites at different genes evolving under positive selection in SARS-CoV-2 lineages affecting cats. Information about the codon sites associated with specific asterisk colors is presented in Figure 5 Black bars label lineages carrying premature stop codons at gen ORF8 codon positions 27 and 64.

**Figure 5 pathogens-12-01361-f005:**
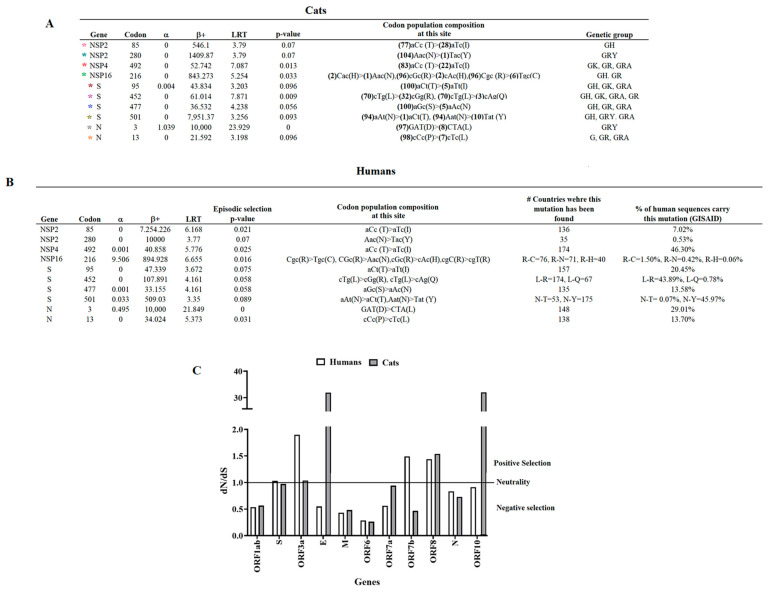
Comparison between the evolutionary dynamics of SARS-CoV-2 lineages that affected cat and human populations during the pandemic. (**A**) A total of 105 sequences were evaluated by MEME analysis to infer potentially relevant codon sites evolving under positive selection in SARS-CoV-2 lineages that affected cat populations during the pandemic. Sites under positive selection were identified with asterisks of different colors (see the tree in Figure 4 to identify lineages displaying specific sites under positive selection). In the column identified as “codon population composition at this site,“ the numbers in parenthesis represent the number of viral sequences in the population carrying this specific codon. The letters in the parenthesis indicate amino acids encoded by specific codons. (**B**) MEME analysis was conducted on 117 viral sequences recovered from humans during the pandemic. Information about the countries where mutations at specific codon sites under positive selection were found and the overall percentage of human sequences carrying these mutations was obtained from the GISAID database. For MEME analysis, α = synonymous substitution rate at the site, β+ = nonsynonymous substitution rate at the site for the positive/neutral evolution component, and LRT = likelihood ratio test statistic for diversifying selection. (**C**) Comparison of dN/dS ratios at different gene segments of SARS-CoV-2 between cat and human populations. Calculations were conducted using MEME.

**Figure 6 pathogens-12-01361-f006:**
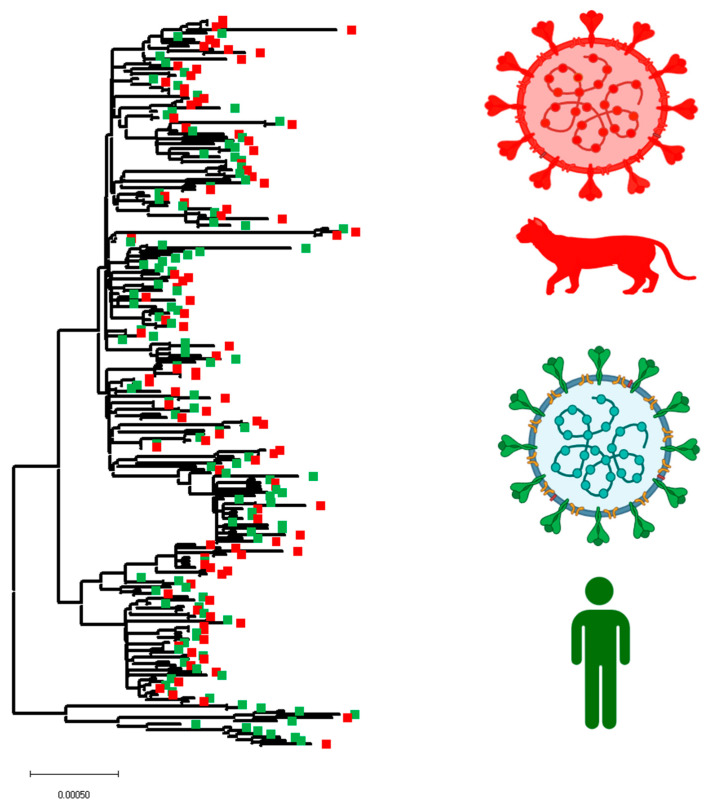
Phylogenomic dynamics between the cat and human SARS-CoV-2 lineages. The phylogenetic analysis was conducted using 222 SARS-CoV-2 lineages (cats n = 105, humans n = 117). The SARS-CoV-2 lineages associated with cats and humans were identified in red and green, respectively. Because of size restrictions, the version of this phylogenetic tree showing the ID of cats and human-recovered viral lineages is shown in Appendix A. This figure was created using BioRender.com under the academic license number WW25XVH794.

**Figure 7 pathogens-12-01361-f007:**
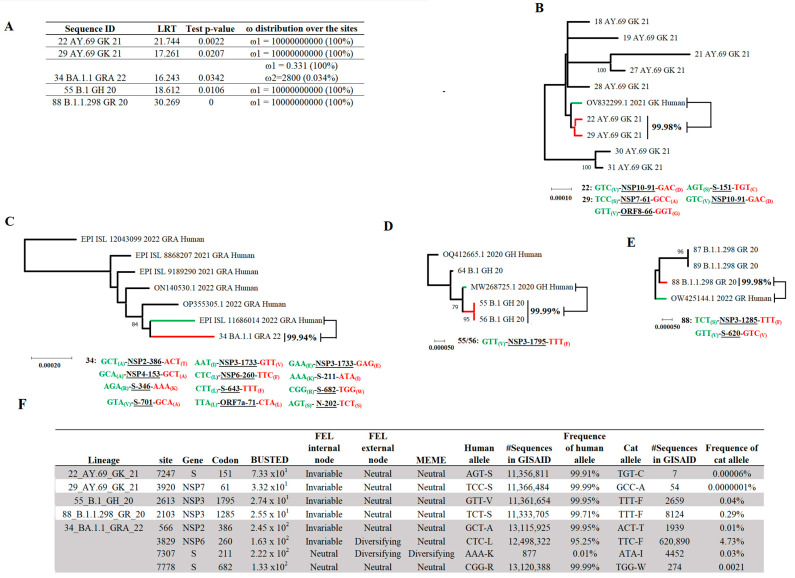
Identification of potential SARS-CoV-2 lineages evolving under positive selection from cat populations. (**A**) Results of the aBSREL analyses identify potential cat lineages evolving because of natural selection. (**B**) Phylogenetic relationship between identified cat SARS-CoV-2 lineages (red branches) 22 AY.69 GK 21, 29 AY.69 GK 21, and its closest related ancestral SARS-CoV-2 human lineage (green branch, sequence OV832299.1 2021 GK human). (**C**) Phylogenetic relationship between identified cat SARS-CoV-2 lineages (red branch) 34 BA.1.1 GRA 22, and its closest related ancestral SARS-CoV-2 human lineage (green branch, sequence EPI ISL 11686014 2022 GRA human). (**D**) Phylogenetic relationship between the identified cat SARS-CoV-2 lineage (red branch) 55 B.1 GH 20 (similar to 56 B.1 GH 20) and its closest related ancestral SARS-CoV-2 human lineage (green branch, sequence MW268725.1 2020 GH human). (**E**) Phylogenetic relationship between the identified cat SARS-CoV-2 lineage (red branches) 88 B.1.1.298 GR 20 and its closest related ancestral SARS-CoV-2 human lineage (green branch, sequence OW425144.1 2022 GR Human). The percentage in bold between cat and human sequences reflects their nucleotide identity. Below, multiple trees display the polymorphic codon sites at different genes impacting the identity between human (information in green) and cat (information in red) sequences. All these short trees were obtained from the phylogenetic analysis in Appendix A. (**F**) Busted analysis was conducted to determine evidence ratios (ER Busted) for specific codon sites associated with the branches linked to the emergence of cat lineages showing ω > 1. The relevance of these sites was assessed by FEL and MEME (cutoff 0.1). FEL analysis was conducted at both internal (selection at the population level) and external (selection at the individual level) nodes. The overall number of sequences in the GISAID database showing the alleles in cat lineages and human ancestral sequences at specific codon positions are shown. Frequencies represented by different alleles in humans and cats at specific codon positions were obtained from the GISAID database. The phylodynamics of multiple codon positions identified by BUSTED analysis are presented in Appendix A.

## Data Availability

No new SARS-CoV-2 sequences were generated during this study. All sequences used for this study are publicly available in the GISAID and GenBank databases.

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
