# Peer review of "Assessing the Potential Role of Cats (Felis catus) as Generators of Relevant SARS-CoV-2 Lineages during the Pandemic"

_pathogens, 2023, doi:10.3390/pathogens12111361_

Round 1
Reviewer 1 Report
Comments and Suggestions for Authors
The study investigates the dynamics and evolutionary characteristics of SARS-CoV-2 infections in cats during the pandemic. Using a dataset of 105 viral genomes from cats and 117 from humans, the research demonstrated a positive correlation between the genetic groups affecting both species, suggesting that cats' viral genetic groups were influenced by human circulation dynamics. The paper's strength is rooted in its comprehensive phylogenomic analysis, which highlights potential independent evolution of the virus in cats and identifies specific lineages and codon sites under positive selection.
However, while the study was conducted solely through in silico methods, the data selection and description require more precision. This work would gain further relevance if complemented by experimental validation.
Comments and Suggestions:
Line 148-150: How were the cat samples selected for the study? Was there any bias in the selection process?
Line 149-153: Are the selected genomes complete? Please provide more details about the length and quality of the genomes used for this study.
line 153: Were there any specific criteria for selecting these 179 reports from the GISAID database?
Line: 178-179: Over 50% of the samples (genomes) were collected from the US, with the remainder representing other countries. Is the evolutionary dynamic of SARS-CoV-2 infecting cats homogeneous across all countries, or are there differences? Clarifying this point will be important to exclude region-dependent factors?
Line 179: Is the higher percentage of sequences from male cats significant? Does it suggest that male cats are more susceptible or were there other factors at play?
line 176: Please provide the unit of age in panel C (figure 01)?
Line 184: In the description provided for Figure 2, the term "affecting" is used to describe the relationship between SARS-CoV-2 viral lineages and various animal species. I suggest considering the term "infecting" if the intention is to highlight the direct transmission and colonization of the virus in these species?
Line 220: How were the 105 sequences selected for the phylogenomic analysis?
Line 241: Regarding the premature stop codons in the ORF8 gene, have similar observations been made in other species or in human populations?
line 259-260: Correct the figure reference to match the manuscript. It should be Figure 6A instead of Figure 5A?
Line 290-303: Given the identified codons under positive selection, are there any known functional implications associated with these specific codons? For instance, do they play a role in virulence, transmissibility, or immune evasion?
lines 290-309: Ensure that the figures referenced align with the content being described “Figure 5A", "Figure 5B", and "Figure 5C". Should these be "Figure 6A", "Figure 6B", and "Figure 6C"?
Line 325-365: Please ensure that the correct figure numbers and panels are consistently referred to throughout the manuscript.
Line 342: Correct the figure reference to match the manuscript. It should be Figure 7 instead of Figure 6
Author Response
We like to thank the reviewer for their time to review our study, and for the valuable comments to improve the content of our manuscript. These are our responses to your comments.
The study investigates the dynamics and evolutionary characteristics of SARS-CoV-2 infections in cats during the pandemic. Using a dataset of 105 viral genomes from cats and 117 from humans, the research demonstrated a positive correlation between the genetic groups affecting both species, suggesting that cats' viral genetic groups were influenced by human circulation dynamics. The paper's strength is rooted in its comprehensive phylogenomic analysis, which highlights potential independent evolution of the virus in cats and identifies specific lineages and codon sites under positive selection.
However, while the study was conducted solely through in silico methods, the data selection and description require more precision. This work would gain further relevance if complemented by experimental validation.
Response: We totally agree with the reviewer. We mentioned this limitation in the discussion section of this study.
Comments and Suggestions:
Line 148-150: How were the cat samples selected for the study? Was there any bias in the selection process?
Response: The criteria of inclusion for samples from cats and other species was the use of all metadata information available at GISAID database at 4/16/2023. It was the time when we started with this study. We highlighted this fact by including more information in sections 2.1 (we included the exact number of reports from each species), 2.2, 3.1 (corrected the number of lineages from all species included in the analysis presented in figure 2), legend of figure 2 was corrected regarding the total number of lineages (n=155)
Line 149-153: Are the selected genomes complete? Please provide more details about the length and quality of the genomes used for this study.
Response: Selected genomes were complete. Information about length and criteria of inclusion regarding to the quality of the sequences was included in section 2.1.
line 153: Were there any specific criteria for selecting these 179 reports from the GISAID database?
Response: No, as stated in the information included in sections 2.1, we included all the reports available in GISAID database at 4/16/2023
Line: 178-179: Over 50% of the samples (genomes) were collected from the US, with the remainder representing other countries. Is the evolutionary dynamic of SARS-CoV-2 infecting cats homogeneous across all countries, or are there differences? Clarifying this point will be important to exclude region-dependent factors?
Response: Great point! Unfortunately, the unequal representation of available viral sequences around the world, represents a main limitation to conduct this analysis. However, we include your valuable observation in the discussion section.
Line 179: Is the higher percentage of sequences from male cats significant? Does it suggest that male cats are more susceptible or were there other factors at play?
Response: Good point. Yes, we performed a Fisher exact test and found a significant correlation, also no correlation was found between reports and gender. Information was included in section 3.1. Also, a statement about the association between number of clinical reports and gender was included in the discussion section. Information in the section 3.1 was improved.
line 176: Please provide the unit of age in panel C (figure 01)?
Response: This figure was corrected.
Line 184: In the description provided for Figure 2, the term "affecting" is used to describe the relationship between SARS-CoV-2 viral lineages and various animal species. I suggest considering the term "infecting" if the intention is to highlight the direct transmission and colonization of the virus in these species?
Response: We agree. It was changed.
Line 220: How were the 105 sequences selected for the phylogenomic analysis?
Response: It was explained in section 2.1 “A total of 105 complete viral genomes from cats naturally infected with SARS-CoV-2 during the pandemic were retrieved from the GISAID database [22]. Average length of sequences was 29,826 nucleotides. These 105 sequences were chosen from a total of 168 full length available sequences under the GISAID criteria of complete (considered genomes >29,000 nt) and high coverage (only entries with <1% nt and <0.05% unique amino acid mutations not seen in other sequences in database, as well and no insertion/deletion unless verified by submitter)”
Line 241: Regarding the premature stop codons in the ORF8 gene, have similar observations been made in other species or in human populations?
Response: Good point. Yes, it has been described in human populations. A statement explaining this result was included in the discussion section.
line 259-260: Correct the figure reference to match the manuscript. It should be Figure 6A instead of Figure 5A?
Response: It was corrected.
Line 290-303: Given the identified codons under positive selection, are there any known functional implications associated with these specific codons? For instance, do they play a role in virulence, transmissibility, or immune evasion?
Response: The function of these codons is informed in the discussion section. “In this context, some of the codon sites predicted under positive selection in both populations, including NSP4- 492 (increased infectivity and evasion of the immune response [35], S-95 (increased infectivity and transmissibility) [36], S-452 (decreased sensitivity to monoclonal antibodies) [37,38,39], S-477 (decreased sensitivity to monoclonal antibodies) [40,41], S-501(decreased sensitivity to monoclonal antibodies, enhance infection and transmission) [42], and N-13 (Affect CD8+ response) [43], have shown their biological relevance for the immune host evasion produced by SARS-CoV-2, suggesting that these positions may provide adaptive advantages for this virus during the infection in both hosts”
lines 290-309: Ensure that the figures referenced align with the content being described “Figure 5A", "Figure 5B", and "Figure 5C". Should these be "Figure 6A", "Figure 6B", and "Figure 6C"?
Response: It was corrected
Line 325-365: Please ensure that the correct figure numbers and panels are consistently referred to throughout the manuscript.
Response: It was corrected.
Line 342: Correct the figure reference to match the manuscript. It should be Figure 7 instead of Figure 6
Response: it was corrected.
Reviewer 2 Report
Comments and Suggestions for Authors
Dear Authors
Thank you very much for the manuscript. Below are suggested comments.
Regards,
Line 38: Please review the website citation. Follow the instructions of the journal.
MDPI indicates “9. Title of Site. Available online: URL (accessed on Day Month Year). Unlike published works, websites may change over time or disappear, so we encourage you create an archive of the cited website using a service such as WebCite. Archived websites should be cited using the link provided as follows: 10. Title of Site. URL (archived on Day Month Year)”.
Line 51: It is suggested to review the following information and, if applicable, update it and include it.
WOAH, 2023. SARS CoV-2 in Animals – Situation Report 22. https://www.woah.org/es/documento/sars-cov-2-in-animals-situation-report-22/
EFSA Panel on Animal Health and Welfare (AHAW), Nielsen, S. S., Alvarez, J., Bicout, D. J., Calistri, P., Canali, E., ... & Ståhl, K. (2023). SARS‐CoV‐2 in animals: susceptibility of animal species, risk for animal and public health, monitoring, prevention and control. EFSA Journal, 21(2), e07822.
Line 62: Please review the website citation.
Line 63: Please include the citation, after “species”.
Introduction: It is suggested to include a paragraph regarding the detections of SARS-CoV-2 in cats. Considering the following references.
Amman, B. R., Cossaboom, C. M., Wendling, N. M., Harvey, R. R., Rettler, H., Taylor, D., ... & Towner, J. S. (2022). GPS Tracking of Free-Roaming Cats (Felis catus) on SARS-CoV-2-Infected Mink Farms in Utah. Viruses, 14(10), 2131.
van Aart, A. E., Velkers, F. C., Fischer, E. A., Broens, E. M., Egberink, H., Zhao, S., ... & Smit, L. A. (2022). SARS‐CoV‐2 infection in cats and dogs in infected mink farms. Transboundary and Emerging Diseases, 69(5), 3001-3007.
Botero, Y., Ramírez, J. D., Serrano-Coll, H., Aleman, A., Ballesteros, N., Martinez, C., ... & Mattar, S. (2022). First report and genome sequencing of SARS-CoV-2 in a cat (Felis catus) in Colombia. Memórias do Instituto Oswaldo Cruz, 117, e210375.
Garigliany, M., Van Laere, A. S., Clercx, C., Giet, D., Escriou, N., Huon, C., ... & Desmecht, D. (2020). SARS-CoV-2 natural transmission from human to cat, Belgium, March 2020. Emerging infectious diseases, 26(12), 3069.
Halfmann, P. J., Hatta, M., Chiba, S., Maemura, T., Fan, S., Takeda, M., ... & Kawaoka, Y. (2020). Transmission of SARS-CoV-2 in domestic cats. New England Journal of Medicine, 383(6), 592-594.
Michelitsch, A., Schön, J., Hoffmann, D., Beer, M., & Wernike, K. (2021). The second wave of SARS-CoV-2 circulation—antibody detection in the domestic cat population in Germany. Viruses, 13(6), 1009.
Mohebali, M., Hassanpour, G., Zainali, M., Gouya, M. M., Khayatzadeh, S., Parsaei, M., ... & Zarei, Z. (2022). SARS-CoV-2 in domestic cats (Felis catus) in the northwest of Iran: Evidence for SARS-CoV-2 circulating between human and cats. Virus Research, 310, 198673.
Musso, N., Costantino, A., La Spina, S., Finocchiaro, A., Andronico, F., Stracquadanio, S., ... & Emmanuele, G. (2020). New SARS-CoV-2 infection detected in an Italian pet cat by RT-qPCR from deep pharyngeal swab. Pathogens, 9(9), 746.
Sailleau, C., Dumarest, M., Vanhomwegen, J., Delaplace, M., Caro, V., Kwasiborski, A., ... & Le Poder, S. (2020). First detection and genome sequencing of SARS‐CoV‐2 in an infected cat in France. Transboundary and emerging diseases, 67(6), 2324-2328.
Segalés, J., Puig, M., Rodon, J., Avila-Nieto, C., Carrillo, J., Cantero, G., ... & Vergara-Alert, J. (2020). Detection of SARS-CoV-2 in a cat owned by a COVID-19− affected patient in Spain. Proceedings of the National Academy of Sciences, 117(40), 24790-24793.
Tewari, D., Boger, L., Brady, S., Livengood, J., Killian, M. L., Nair, M. S., ... & Brightbill, K. (2022). Transmission of SARS‐CoV‐2 from humans to a 16‐year‐old domestic cat with comorbidities in Pennsylvania, USA. Veterinary Medicine and Science, 8(2), 899-906.
Villanueva‐Saz, S., Giner, J., Tobajas, A. P., Pérez, M. D., González‐Ramírez, A. M., Macías‐León, J., ... & Fernández, A. (2022). Serological evidence of SARS‐CoV‐2 and co‐infections in stray cats in Spain. Transboundary and Emerging Diseases, 69(3), 1056-1064.
Line 103: Could you include the criteria for sequence inclusion (quality)?
Line 108: Could you please clarify the number of sequences per species of other species included in the study? It is suggested to include in the materials and methods description.
A total of 105 complete viral genomes from cats naturally infected with SARS-CoV-2 during the pandemic were retrieved from the GISAID database. A total of 179 SARS-CoV-2 were analyzed by hierarchical cluster analysis. Additionally, 117 complete viral genomes from human cases showing the highest levels of identity against cat SARS-CoV-107 2 genomes were obtained from the GenBank database. Other species?
Line 198: Please explain the use of lineages (PANGO lineages) for the analysis in Figure 2 and genetic groups for Figure 3 (GISAIS Clades). It is suggested to explain in the methodology.
Line 220: It is suggested to include information on sequence inclusion criteria and quality in the materials and methods section. Describe the total available sequences and the total selected sequences (%).
Line 256 to 283: It is suggested to include this in materials and methods. Limit this paragraph to results.
Line 389. Include references corresponding to each statement included in the paragraph.
Line 47: Please describe and include references.
Line 437: Please include references (describe how the sample date influences the frequency of strains by species).
Line 457: Could you support with references?
Line 530: It suggested to standardize the use of variant nomenclature.
Author Response
We like to thank the reviewer for their time to review our study, and for the valuable comments to improve the content of our manuscript. These are our responses to your comments.
Line 38: Please review the website citation. Follow the instructions of the journal.
Response= It was corrected.
MDPI indicates “9. Title of Site. Available online: URL (accessed on Day Month Year). Unlike published works, websites may change over time or disappear, so we encourage you create an archive of the cited website using a service such as WebCite. Archived websites should be cited using the link provided as follows: 10. Title of Site. URL (archived on Day Month Year)”.
Line 51: It is suggested to review the following information and, if applicable, update it and include it.
Response: Information was reviewed and included in the text.
WOAH, 2023. SARS CoV-2 in Animals – Situation Report 22. https://www.woah.org/es/documento/sars-cov-2-in-animals-situation-report-22/
EFSA Panel on Animal Health and Welfare (AHAW), Nielsen, S. S., Alvarez, J., Bicout, D. J., Calistri, P., Canali, E., ... & Ståhl, K. (2023). SARS‐CoV‐2 in animals: susceptibility of animal species, risk for animal and public health, monitoring, prevention and control. EFSA Journal, 21(2), e07822.
Line 62: Please review the website citation.
Response: it was corrected
Line 63: Please include the citation, after “species”.
Response: It was included.
Introduction: It is suggested to include a paragraph regarding the detections of SARS-CoV-2 in cats. Considering the following references.
Response: Thank you for providing these valuable references. We reviewed all of them and include some statements to enhance the introduction section of our study.
Amman, B. R., Cossaboom, C. M., Wendling, N. M., Harvey, R. R., Rettler, H., Taylor, D., ... & Towner, J. S. (2022). GPS Tracking of Free-Roaming Cats (Felis catus) on SARS-CoV-2-Infected Mink Farms in Utah. Viruses, 14(10), 2131.
van Aart, A. E., Velkers, F. C., Fischer, E. A., Broens, E. M., Egberink, H., Zhao, S., ... & Smit, L. A. (2022). SARS‐CoV‐2 infection in cats and dogs in infected mink farms. Transboundary and Emerging Diseases, 69(5), 3001-3007.
Botero, Y., Ramírez, J. D., Serrano-Coll, H., Aleman, A., Ballesteros, N., Martinez, C., ... & Mattar, S. (2022). First report and genome sequencing of SARS-CoV-2 in a cat (Felis catus) in Colombia. Memórias do Instituto Oswaldo Cruz, 117, e210375.
Garigliany, M., Van Laere, A. S., Clercx, C., Giet, D., Escriou, N., Huon, C., ... & Desmecht, D. (2020). SARS-CoV-2 natural transmission from human to cat, Belgium, March 2020. Emerging infectious diseases, 26(12), 3069.
Halfmann, P. J., Hatta, M., Chiba, S., Maemura, T., Fan, S., Takeda, M., ... & Kawaoka, Y. (2020). Transmission of SARS-CoV-2 in domestic cats. New England Journal of Medicine, 383(6), 592-594.
Michelitsch, A., Schön, J., Hoffmann, D., Beer, M., & Wernike, K. (2021). The second wave of SARS-CoV-2 circulation—antibody detection in the domestic cat population in Germany. Viruses, 13(6), 1009.
Mohebali, M., Hassanpour, G., Zainali, M., Gouya, M. M., Khayatzadeh, S., Parsaei, M., ... & Zarei, Z. (2022). SARS-CoV-2 in domestic cats (Felis catus) in the northwest of Iran: Evidence for SARS-CoV-2 circulating between human and cats. Virus Research, 310, 198673.
Musso, N., Costantino, A., La Spina, S., Finocchiaro, A., Andronico, F., Stracquadanio, S., ... & Emmanuele, G. (2020). New SARS-CoV-2 infection detected in an Italian pet cat by RT-qPCR from deep pharyngeal swab. Pathogens, 9(9), 746.
Sailleau, C., Dumarest, M., Vanhomwegen, J., Delaplace, M., Caro, V., Kwasiborski, A., ... & Le Poder, S. (2020). First detection and genome sequencing of SARS‐CoV‐2 in an infected cat in France. Transboundary and emerging diseases, 67(6), 2324-2328.
Segalés, J., Puig, M., Rodon, J., Avila-Nieto, C., Carrillo, J., Cantero, G., ... & Vergara-Alert, J. (2020). Detection of SARS-CoV-2 in a cat owned by a COVID-19− affected patient in Spain. Proceedings of the National Academy of Sciences, 117(40), 24790-24793.
Tewari, D., Boger, L., Brady, S., Livengood, J., Killian, M. L., Nair, M. S., ... & Brightbill, K. (2022). Transmission of SARS‐CoV‐2 from humans to a 16‐year‐old domestic cat with comorbidities in Pennsylvania, USA. Veterinary Medicine and Science, 8(2), 899-906.
Villanueva‐Saz, S., Giner, J., Tobajas, A. P., Pérez, M. D., González‐Ramírez, A. M., Macías‐León, J., ... & Fernández, A. (2022). Serological evidence of SARS‐CoV‐2 and co‐infections in stray cats in Spain. Transboundary and Emerging Diseases, 69(3), 1056-1064.
Line 103: Could you include the criteria for sequence inclusion (quality)?
Response: The information was included in section 2.1. “A total of 105 complete viral genomes from cats naturally infected with SARS-CoV-2 during the pandemic were retrieved from the GISAID database [22]. Average length of sequences was 29,826 nucleotides. These 105 sequences were chosen from a total of 168 full length available sequences under the GISAID criteria of complete (considered genomes >29,000 nt) and high coverage (only entries with <1% nt and <0.05% unique amino acid mutations not seen in other sequences in database, as well and no insertion/deletion unless verified by submitter)”
Line 108: Could you please clarify the number of sequences per species of other species included in the study? It is suggested to include in the materials and methods description.
A total of 105 complete viral genomes from cats naturally infected with SARS-CoV-2 during the pandemic were retrieved from the GISAID database. A total of 179 SARS-CoV-2 were analyzed by hierarchical cluster analysis. Additionally, 117 complete viral genomes from human cases showing the highest levels of identity against cat SARS-CoV-107 2 genomes were obtained from the GenBank database. Other species?
Response= We apologize for the confusion about it. We clarified it in section 2.1. Overall, for phylogenetic and evolutionary analyses a total of 105 sequences from cats and 117 from humans were used. On the other hand, for hierarchical cluster analysis, we used information for the reports regarding the proportion of cases associated with the infection of specific pangolin lineages. In this sense, in section 2.1, we included the number of reports from each species used in this study. Also, sections 2.2 and 3.2, both related to this analysis were updated.
Line 198: Please explain the use of lineages (PANGO lineages) for the analysis in Figure 2 and genetic groups for Figure 3 (GISAIS Clades). It is suggested to explain in the methodology.
Response= It was explained in sections 2.2 , and 2.3.
Line 220: It is suggested to include information on sequence inclusion criteria and quality in the materials and methods section. Describe the total available sequences and the total selected sequences (%).
Response: this information was included in section 2.1
Line 256 to 283: It is suggested to include this in materials and methods. Limit this paragraph to results.
Response: It was corrected.
Line 389. Include references corresponding to each statement included in the paragraph.
Response: It was corrected.
Line 47: Please describe and include references.
Response: It was corrected.
Line 437: Please include references (describe how the sample date influences the frequency of strains by species).
Response: It was included in the discussion section.
Line 457: Could you support with references?
Response: reference was included.
Line 530: It suggested to standardize the use of variant nomenclature.
Response: It was corrected